# LAYER-WISE FEEDBACK SIGNALS: DYNAMIC REGULATION FOR CONTINUAL LEARNING

## ABSTRACT

Continual learning aims to acquire new tasks while preserving performance on previously learned ones, but most methods struggle with catastrophic forgetting. Existing approaches typically treat all layers uniformly, often trading stability for plasticity or vice versa. However, different layers naturally exhibit varying levels of uncertainty (entropy) when classifying tasks. High-entropy layers tend to underfit by failing to capture task-specific patterns, while low-entropy layers risk overfitting by becoming overly confident and specialized. To address this imbalance, we propose an entropy-aware continual learning method that employs a dynamic feedback mechanism to regulate each layer based on its entropy. Specifically, our approach reduces entropy in high-entropy layers to mitigate underfitting and increases entropy in overly confident layers to alleviate overfitting. This adaptive regulation encourages the model to converge to wider local minima, which have been shown to improve generalization. Our method is general and can be seamlessly integrated with both replay- and regularization-based approaches. Experiments on Split-CIFAR100 and Tiny-ImageNet demonstrate substantial performance gains over state-of-the-art baselines.

## 1 INTRODUCTION

Continual learning (Masana et al., 2023; De Lange et al., 2022) aims to enable models to acquire new tasks without suffering from catastrophic forgetting (McCloskey & Cohen, 1989; French, 1999), the degradation of performance on previously learned knowledge. This issue stems from the fundamental stability-plasticity dilemma; a model must be stable enough to preserve old knowledge while remaining sufficiently plastic to learn new information. To navigate this trade-off, research has predominantly explored three families of approaches: regularization-based, replay-based, and parameter-expansion methods.

A primary limitation of many existing approaches is their reliance on simple regularization techniques, such as L1 or L2, which often guide the model toward sharp minima in the loss landscape, which are known to generalize poorly (Foret et al., 2021). This convergence to a narrow "deep well" rather than a broad "flat valley" is a form of overfitting that degrades test-time performance. Furthermore, these methods are typically layer-agnostic, lacking any mechanism to modulate learning based on layer-specific performance. By failing to preserve the knowledge in well-performing layers while simultaneously encouraging underperforming layers to adapt, they compromise the model's ability to balance stability and plasticity, ultimately hindering overall accuracy.

Our primary contribution is a novel technique we call **Self-Adaptive Entropy Scaling**. While regularizing the final layer's classification entropy is a known and effective technique for improving model robustness (Cha et al., 2021b), a naive, uniform application to all layers has a significant drawback. Indiscriminately penalizing layers that already exhibit high entropy can be detrimental, potentially degrading valuable learned representations. To address this, our entropy scaling method adaptively adjusts the regularization strength for each layer through the lens of Bayesian inference. The penalty is applied strongly to layers with low entropy (i.e., those with over-confident outputs) while being reduced for layers that already possess high-entropy features, preserving their diversity.

To further enhance the efficacy of entropy scaling, we introduce a complementary adaptive training mechanism. Our adaptive training modulates the plasticity of each layer based on its performance on previous tasks. Specifically, we constrain updates for high-performing layers to preserve their

acquired knowledge, while conversely amplifying updates for underperforming layers to encourage more rapid adaptation.

To demonstrate the effectiveness of our approach, we conduct both theoretical analysis and empirical evaluation. Theoretically, we show that our method leads to a tighter generalization error bound. Empirically, our experiments on standard image classification benchmarks confirm that the proposed approach significantly improves average accuracy while simultaneously reducing forgetting, outperforming state-of-the-art methods. Our contributions are as follows:

- We propose a novel framework for continual learning that employs a dynamic feedback mechanism to apply layer-aware regularization, overcoming the limitations of layer-agnostic approaches.

- We design a new algorithm that integrates two techniques, entropy scaling and adaptive training through Bayesian inference, to intelligently modulate plasticity across the network.

- We conduct in-depth theoretical analysis that firmly supports the effectiveness of our method.

- We conduct comprehensive experiments on popular continual learning datasets, achieving state-of-the-art results and showing marked improvements in both accuracy and knowledge retention.

## 2 RELATED WORK

Continual learning (CL) addresses the challenge of training models on a sequence of tasks without catastrophically forgetting previously acquired knowledge. To this end, three primary classes of methods have been developed. Regularization-based approaches (Rebuffi et al., 2017; Zenke et al., 2017; Nguyen et al., 2018; Aljundi et al., 2018; Yan et al., 2024) introduce penalty terms into the loss function to constrain updates on parameters critical for past tasks. Another line of work, memory-replay, maintains a buffer of exemplars from previous tasks (Shin et al., 2017; Rolnick et al., 2019; Lopez-Paz & Ranzato, 2017; Riemer et al., 2018; Pham et al., 2021; Arani et al., 2022; Verwimp et al., 2021) that are revisited during subsequent training to prevent knowledge degradation. A third approach, architecture expansion (Rusu et al., 2022; Mallya & Lazebnik, 2018; Serra et al., 2018; Li et al., 2019; Hung et al., 2019), dynamically grows the network by adding new weights or adapters as new tasks arrive, thereby isolating task-specific parameters to prevent interference.

While these general strategies are effective, recent works have explored output regularization-based approaches. For instance, CPR (Cha et al., 2021b) provides a strong baseline by regulating only the final output layer to find wider local minima. However, this singular focus means that the crucial intermediate layers are not explicitly regularized, potentially limiting their robustness against forgetting. In our work, we argue that not all layers are created equal. We bridge this gap by introducing a method that dynamically applies supervision across multiple layers, recognizing that earlier and later layers play distinct roles. This allows our model to reap the benefits of broad minima across the entire network, unlike CPR, while also leveraging the unique contributions of each layer.

## 3 METHOD

### 3.1 PRELIMINARIES

In the standard continual learning (CL) setup, a model is trained on a sequence of tasks, arriving one after another. Let the sequence of tasks be denoted by $\mathcal{T} = \{\mathcal{T}_1, \mathcal{T}_2, \ldots, \mathcal{T}_N\}$, where $N$ is the total number of tasks. Each task $\mathcal{T}_t$ for $t \in \{1, \ldots, N\}$ is associated with its own data distribution $\mathcal{D}_t = \{(\mathbf{x}_i, y_i)\}$, where $\mathbf{x}_i$ represents the input data and $y_i$ is the corresponding label. Except for a small memory buffer, the data from previous tasks $\mathcal{T}_1, \ldots, \mathcal{T}_{t-1}$ is not available when the model is learning the current task $\mathcal{T}_t$.

Our model is represented by a function $f(\cdot; \boldsymbol{\theta})$, parameterized by a set of parameters $\boldsymbol{\theta} \in \mathbb{R}^d$. The goal of the model is to learn a mapping from inputs to outputs. Upon observing task $\mathcal{T}_t$, the model updates its parameters $\boldsymbol{\theta}$ to minimize a task-specific loss function, $\mathcal{L}_t$. This loss is typically

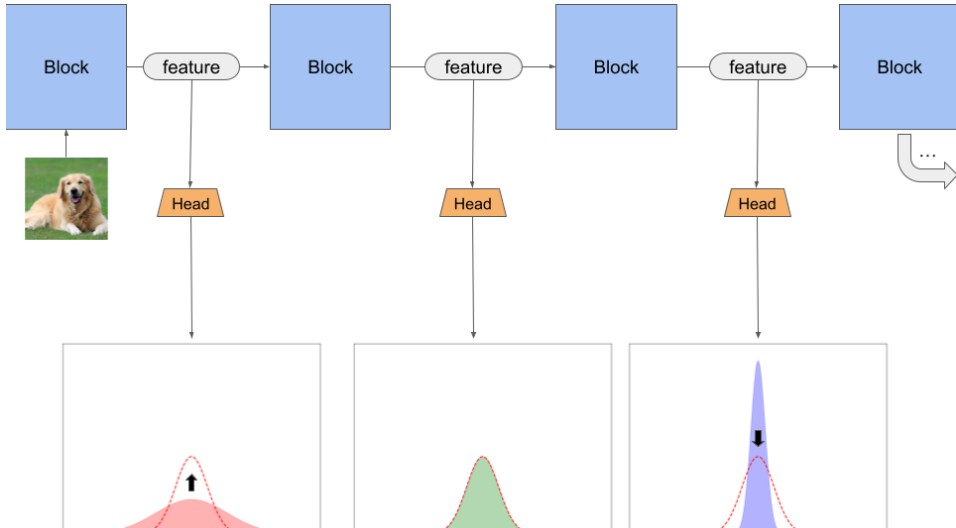

Figure 1: As the input data propagates through each successive block of the network, its feature vector at each layer is fed into a dedicated classification head. We calculate the entropy of the resulting output distribution from each head. This entropy measurement then dynamically adjusts the strength of the regularization term applied to that specific layer, allowing for adaptive regularization throughout the model. Without our method, layer output entropies (solid red, green, and blue) exhibit high variance, indicating that some layers become over-confident while others remain uncertain. Our method guides the entropy of each layer towards a stable, medium-entropy target (dashed red line), promoting more consistent representations throughout the network.

computed as the empirical risk over the data distribution $\mathcal{D}_t$:

$$\mathcal{L}_t(\boldsymbol{\theta}) = \mathbb{E}_{(\mathbf{x},y) \sim \mathcal{D}_t}[\ell(f(\mathbf{x}; \boldsymbol{\theta}), y)] \tag{1}$$

where $\ell(\cdot, \cdot)$ is a standard loss function, such as cross-entropy for classification. Let $\boldsymbol{\theta}_{t-1}^*$ denote the optimal parameters found after training on tasks up to $\mathcal{T}_{t-1}$. When task $\mathcal{T}_t$ arrives, the learning process aims to find a new set of parameters $\boldsymbol{\theta}_t^*$ that minimizes $\mathcal{L}_t(\boldsymbol{\theta})$ without significantly increasing the loss on previous tasks. This is the core challenge of continual learning, known as catastrophic forgetting. The ideal objective of a continual learning agent is to find a single set of parameters $\boldsymbol{\theta}_N^*$ that performs well across all tasks simultaneously. This can be formulated as minimizing the total loss over the entire sequence:

$$\boldsymbol{\theta}_N^* = \arg \min_{\boldsymbol{\theta}} \sum_{t=1}^{N} \mathcal{L}_t(\boldsymbol{\theta}) \tag{2}$$

However, due to the sequential and constrained nature of data availability, achieving this joint optimization directly is not feasible. The primary goal of CL methods is to approximate this solution by sequentially updating the parameters $\boldsymbol{\theta}$ in a way that balances performance on the current task with the preservation of knowledge from past tasks.

## 3.2 GUIDED ENTROPY-ADAPTIVE FEEDBACK FOR CONTINUAL LEARNING (GRACE)

Our proposed method, GRACE, is a framework designed to mitigate catastrophic forgetting in continual learning. The fundamental principle is to dynamically adjust the learning process for each layer based on two key signals: its current output entropy and its historical performance on past tasks. We formulate this as a general optimization problem where we adaptively scale the standard task and augment it with a regularization term that is entropy-scaled.

The general optimization objective for a given task is:

$$\mathcal{L}(\boldsymbol{\theta}) = \mathcal{L}_t(\boldsymbol{\theta}) + R_t(\boldsymbol{\theta}) = \sum_{l=1}^{L} (\alpha_l \cdot \mathcal{L}_l(\boldsymbol{\theta}) + \gamma_l \cdot R_l(\boldsymbol{\theta})) \tag{3}$$

Here, $\mathcal{L}_l$ represents the primary objective function cross-entropy loss of layer $l$ on the current task, and it is modified by $\alpha_l$, an adaptive training modulator that scales the main loss term based on the layer's historical performance. For each layer, the regularization function $R_l$ is modulated by $\gamma_l$, an entropy scaling factor. We sum over all the layers $1 \ldots L$ to get the total task loss $\mathcal{L}_t$. The following sections detail the two core components of this framework: entropy scaling and adaptive training, which respectively define $\gamma_l$ and $\alpha_l$.

---

**Algorithm 1** Training with Layer-wise Adaptive Regularization

---

**Require:** ResNet model $f(\cdot; \boldsymbol{\theta})$ with $L$ layers. Sequence of training datasets $D_1, \ldots, D_T$. Validation set $D_{val}$. Learning rate $\eta$.

1: Initialize modulators $\alpha_l \leftarrow 1$ for all $l \in \{1, \ldots, L\}$.
2: **for** task $t \leftarrow 1$ to $T$ **do**               $\triangleright$ Adaptive Training: Calculate $\alpha_l$ for the current task
3:     **if** $t > 1$ **then**
4:         Let $\mathcal{A}_{\text{set}}$ be an empty set
5:         **for** $l \leftarrow 1$ to $L$ **do**
6:             $A_l \leftarrow \text{EvaluateAccuracy}(f(\cdot; \boldsymbol{\theta})_l, D_{val})$          $\triangleright$ Evaluate layer on past data
7:             Add $A_l$ to $\mathcal{A}_{\text{set}}$
8:         **end for**
9:         $\mu_A \leftarrow \text{Mean}(\mathcal{A}_{\text{set}})$    $\sigma_A \leftarrow \text{StdDev}(\mathcal{A}_{\text{set}})$    $\mathcal{L}_t \leftarrow 0$
10:        **for** $l \leftarrow 1$ to $L$ **do**
11:            $s_l \leftarrow (A_l - \mu_A)/\sigma_A$
12:            $\alpha_l \leftarrow e^{\tanh(-s_l)}$                 $\triangleright$ Calculate the modulator based on the score
13:            $\mathcal{L}_t \leftarrow \mathcal{L}_t + \alpha_l \cdot \mathcal{L}_l$
14:        **end for**
15:     **end if**
16:     **for** each training epoch **do**                               $\triangleright$ Train on current task $t$
17:         **for** each mini-batch $(X_b, Y_b) \in D_t$ **do**
18:             Perform forward pass to get activations $h_l$ for each layer $l$.
19:             Let $\mathcal{H}_{\text{set}}$ be an empty list
20:             **for** $l \leftarrow 1$ to $L$ **do**
21:                 $p_l \leftarrow \text{softmax}(h_l)$,    $H(p_l) \leftarrow -\sum_i p_{l,i} \log p_{l,i}$
22:                 $\bar{H}_l \leftarrow \mathbb{E}_{(\mathbf{x},y) \in (X_b, Y_b)}[H(p_l)]$               $\triangleright$ Average entropy over batch
23:                 Add $\bar{H}_l$ to $\mathcal{H}_{\text{set}}$
24:             **end for**
25:             $\mu_H \leftarrow \text{Mean}(\mathcal{H}_{\text{set}}), \sigma_H \leftarrow \text{StdDev}(\mathcal{H}_{\text{set}}), \mathcal{R}_t \leftarrow 0$
26:             **for** $l \leftarrow 1$ to $L$ **do**
27:                 $z_l \leftarrow (\mathcal{H}_l - \mu_H)/\sigma_H$
28:                 $\gamma_l \leftarrow e^{\tanh(z_l)}$               $\triangleright$ Calculate scaling factor based on z-score
29:                 $\mathcal{R}_t \leftarrow \mathcal{R}_t + \gamma_l \cdot \bar{H}_l$
30:             **end for**
31:             $\mathcal{L} \leftarrow \mathcal{L}_t + \mathcal{R}_t$
32:             Update parameters: $\boldsymbol{\theta} \leftarrow \boldsymbol{\theta} - \eta \nabla_{\boldsymbol{\theta}} \mathcal{L}$.
33:         **end for**
34:     **end for**
35:     Update $D_{val}$ with representative samples from current task $t$.
36: **end for**

---

**Self-Adaptive Entropy Scaling** ($\gamma_l$): The goal of entropy scaling is to encourage layers with over-confident (low-entropy) outputs to learn more generalizable representations, while protecting the features of layers that already exhibit high-entropy, diverse outputs. To achieve this, our method dynamically modulates the regularization penalty based on a layer's relative entropy compared to other layers within the same mini-batch, rather than its absolute entropy.

We propose a principled adaptive entropy scaling approach. We model the adaptive entropy scaling factor $\gamma_\ell$ in layer $\ell$ as a latent variable in a Bayesian framework. The goal is to infer a suitable regularization strength for each layer, based on its entropy $H_\ell$. We use variational inference to approximate the posterior distribution over $\gamma_\ell$ given the observed entropy.

Let $\gamma_\ell \in \mathbb{R}_+$ be the entropy regularization strength for layer $\ell$, and $H_\ell \in \mathbb{R}_+$ the entropy observed at that layer. We assume a Gaussian noise model for entropy given regularization:

$$H_\ell = H^* + \frac{c}{\gamma_\ell} + \varepsilon_\ell, \quad \varepsilon_\ell \sim \mathcal{N}(0, \sigma^2)$$

Where $H^*$ is a target entropy value – a reference or anchor that represents the desired entropy level for a model layer. We do not need to estimate $H^*$ since our approach can normalize the entropy values across different layers without knowing $H^*$. $c$ is a constant and $\sigma$ is the standard deviation. Thus, the likelihood becomes:

$$p(H_\ell \mid \gamma_\ell) = \mathcal{N}\left( H_\ell \mid H^* + \frac{c}{\gamma_\ell}, \sigma^2 \right)$$

We place a log-normal prior on $\gamma_\ell$:

$$p(\gamma_\ell) = \text{LogNormal}(\mu_0, \tau^2) \quad \Rightarrow \quad \log \gamma_\ell \sim \mathcal{N}(\mu_0, \tau^2)$$

A log-normal prior is chosen for $\gamma_\ell$ because it ensures positivity, as $\gamma_\ell > 0$ by design. It naturally models multiplicative uncertainty, which is appropriate for scaling factors in entropy regularization. The distribution also has heavy tails, allowing the model to flexibly assign both strong and weak regularization across layers. Furthermore, operating in log-space– where $\log \gamma_\ell \sim \mathcal{N}$– enables efficient variational inference via the reparameterization trick and permits closed-form KL divergence computation. The posterior over $\gamma_\ell$ is intractable, so we approximate it via variational inference. Let the variational posterior be:

$$q_\phi(\gamma_\ell) = \text{LogNormal}(\mu_\phi, \sigma_\phi^2) \quad \Rightarrow \quad \log \gamma_\ell \sim \mathcal{N}(\mu_\phi, \sigma_\phi^2)$$

We optimize the evidence lower bound (ELBO):

$$\mathcal{L}(\phi) = \mathbb{E}_{\gamma_\ell \sim q_\phi} \left[ \log p(H_\ell \mid \gamma_\ell) + \log p(\gamma_\ell) - \log q_\phi(\gamma_\ell) \right]$$

Therefore, the ELBO becomes:

$$\mathcal{L}(\phi) = \mathbb{E}_{\gamma_\ell \sim q_\phi} \Bigg[ -\frac{1}{2\sigma^2} \left( H_\ell - H^* - \frac{c}{\gamma_\ell} \right)^2 - \frac{1}{2\tau^2} (\log \gamma_\ell - \mu_0)^2 - \log \gamma_\ell$$
$$+ \frac{1}{2\sigma_\phi^2} (\log \gamma_\ell - \mu_\phi)^2 + \log \gamma_\ell \Bigg] + \text{const}$$

To avoid optimizing $\mathcal{L}(\phi)$ explicitly during training, we approximate the *posterior mean*:

$$\hat{\gamma}_\ell = \mathbb{E}_{q_\phi}[\gamma_\ell] = \exp\left( \mu_\phi + \frac{\sigma_\phi^2}{2} \right) \tag{4}$$

Assuming a small variance $\sigma_\phi^2 \approx 0$, we approximate $\hat{\gamma}_\ell \approx \exp(\mu_\phi)$. Empirically, we set:

$$\mu_\phi \approx \tanh(z_\ell), \quad \text{where } z_\ell = \frac{H_\ell - \mu_H}{\sigma_H} \tag{5}$$

For each mini-batch, we first compute the average output entropy $\bar{H}_l$ for every layer $l$. The $\mu_H$ denotes the mean of this set of entropies $\{\bar{H}_1, \ldots, \bar{H}_L\}$ and $\sigma_H$ denotes the corresponding standard deviation. $z_\ell$ (z-score) denotes the relative entropy for each layer. This z-score, which measures how far a layer's entropy deviates from the batch average, is used to compute the final scaling factor $\gamma_l$. This yields the final approximation used in GRACE:

$$\gamma_\ell \approx \exp\left( \tanh\left( \frac{H_\ell - \mu_H}{\sigma_H} \right) \right) \tag{6}$$

This formulation ensures an inverse relationship between relative entropy and the regularization strength. A layer with lower-than-average entropy (negative $z_l$) will yield a scaling factor $\gamma_l < 1$, promoting higher entropy to reduce overconfident predictions. Conversely, a layer with higher-than-average entropy (positive $z_l$) will receive a scaling factor $\gamma_l > 1$, thereby strengthening the regularization effect to reduce entropy and alleviate underfitting.

**Adaptive Training** ($\alpha_l$): Inspired by our proposed Self-Adaptive Entropy Scaling, the principle of adaptive training is to dynamically adjust each layer's plasticity based on its performance on previously seen tasks. This allows us to preserve knowledge in stable, well-performing layers while encouraging adaptation in underperforming ones. This is implemented via the learning modulator $\alpha_l$. After completing training on a task, we evaluate the average accuracy $A_l$ for each layer on a validation set of past tasks. We then quantify the relative performance of each layer by calculating its z-score, which measures the deviation from the mean accuracy ($\mu_A$) in units of standard deviation ($\sigma_A$). (Here we use the variable $s$ for "score" to differentiate it from the z-score used in entropy scaling): $s_l = \frac{A_l - \mu_A}{\sigma_A}$. This z-score is then mapped to the modulator $\alpha_l$ for the *next* training task. This mapping is designed such that high-performing layers ($s_l > 0$) receive a smaller $\alpha_l$ (e.g., $< 1$), reducing the impact of regularization and thus preserving their weights. Conversely, underperforming layers ($s_l < 0$) receive a larger $\alpha_l$ (e.g., $> 1$), emphasizing their importance. Finally, we calculate $\alpha_l$ as follows, to bound it within a reasonable range around 1:

$$\alpha_l = e^{\tanh(-s_l)} \tag{7}$$

We use $\alpha_l$ to modify this $\mathcal{L}_l$ (the loss for the classification head attached to layer $l$ on the current task's data). We then present the detailed algorithm in Algorithm 1:

## 4 THEORETICAL ANALYSIS

We perform theoretical analysis about the generalization error with our adaptive entropy control in Theorem 4.4 and forgetting bound in Theorem 4.5.

**Assumption 4.1.** (Smoothness). Each population objective $\mathcal{L}_t$ is $L_t$-smooth; i.e., for all $\boldsymbol{\theta}, \boldsymbol{\theta}'$,

$$\left\| \nabla \mathcal{L}_t(\boldsymbol{\theta}) - \nabla \mathcal{L}_t(\boldsymbol{\theta}') \right\| \leq L_t \left\| \boldsymbol{\theta} - \boldsymbol{\theta}' \right\|.$$

**Assumption 4.2.** (Entropy Lipschitzness). Along the training trajectory, each layer-entropy is Lipschitz in the parameters: for all $\boldsymbol{\theta}, \boldsymbol{\theta}'$ and each layer $\ell$,

$$|H_l(\boldsymbol{\theta}) - H_l(\boldsymbol{\theta}')| \leq c_\ell \left\| \boldsymbol{\theta} - \boldsymbol{\theta}' \right\|.$$

**Assumption 4.3.** (Posterior concentration). The training algorithm induces a posterior $q_t$ with finite second moment such that

$$\mathbb{E}_{\boldsymbol{\theta} \sim q_t} \left\| \boldsymbol{\theta} - \bar{\boldsymbol{\theta}}_t \right\|^2 \leq \sigma_t^2,$$

where $\bar{\boldsymbol{\theta}}_t := \mathbb{E}_{\boldsymbol{\theta} \sim q_t}[\boldsymbol{\theta}]$.

**Cumulative entropy deviation.** We define the cumulative (layerwise) entropy deviation at task $t$:

$$\Delta_t := \sum_{\ell=1}^{L} \left( H_l(\boldsymbol{\theta}_t) - H_{\ell,t}^* \right)^2.$$

where $H_{\ell,t}^*$ denotes the target entropy for layer $l$ at task $t$.

**Theorem 4.4** (PAC-Bayes generalization with entropy control)**.** *Fix $\delta \in (0,1)$. For task $t$ with sample size $n_t \geq 2$, let*

$$\mathcal{L}_t(\boldsymbol{\theta}) = R_t(\boldsymbol{\theta}) + \lambda \Delta_t(\boldsymbol{\theta}), \qquad R_t(\boldsymbol{\theta}) = \mathbb{E}_{(x,y) \sim \mathcal{D}_t}[\ell_t(f_{\boldsymbol{\theta}}(x), y)],$$

*where $\ell_t \in [0,1]$ and $\Delta_t(\boldsymbol{\theta}) = \sum_{\ell=1}^{L} \left( H_l(\boldsymbol{\theta}) \right) - H_{\ell,t}^* \big)^2$.*

*Let $\hat{\mathcal{L}}_t(\boldsymbol{\theta}) = \hat{R}_t(\boldsymbol{\theta}) + \lambda \hat{\Delta}_t(\boldsymbol{\theta})$ be the empirical analogue on $n_t$ samples. For any posterior $q_t$ absolutely continuous w.r.t. a prior $p_t$ (chosen before seeing the task-$t$ data), with probability at least $1 - \delta$ over the sample,*

$$\mathbb{E}_{\boldsymbol{\theta} \sim q_t} \left[ \mathcal{L}_t(\boldsymbol{\theta}) \right] \leq \mathbb{E}_{\boldsymbol{\theta} \sim q_t} \left[ \hat{\mathcal{L}}_t(\boldsymbol{\theta}) \right] + \sqrt{\frac{\mathrm{KL}(q_t \| p_t) + \ln \frac{2\sqrt{n_t}}{\delta}}{2(n_t - 1)}} + \lambda \mathbb{E}_{\boldsymbol{\theta} \sim q_t} \left[ \Delta_t(\boldsymbol{\theta}) \right]. \tag{8}$$

*Moreover, if $p_t = \mathcal{N}(\boldsymbol{\theta}_{t-1}, \Sigma_p)$ and $q_t$ has mean $\bar{\boldsymbol{\theta}}_t$ and covariance $\Sigma_q$,*

$$\mathrm{KL}(q_t \| p_t) \leq \frac{\kappa_t}{2} \| \bar{\boldsymbol{\theta}}_t - \boldsymbol{\theta}_{t-1} \|^2 + C_t, \qquad \kappa_t := \lambda_{\max}(\Sigma_p^{-1}), \quad C_t := \tfrac{1}{2} \left( \mathrm{tr}(\Sigma_p^{-1} \Sigma_q) - k + \ln \frac{\det \Sigma_p}{\det \Sigma_q} \right).$$

**Theorem 4.5** (Forgetting bound via parameter drift). *Let $\mathcal{F}_{s\to t} := \mathcal{L}_s(\boldsymbol{\theta}_t) - \mathcal{L}_s(\boldsymbol{\theta}_s)$ for $1 \leq s < t \leq T$. Assume:*

*(A1) $\ell_s \in [0,1]$ and along the optimization trajectory the population objective $\mathcal{L}_s$ has bounded gradient: $\sup_{\boldsymbol{\theta}\in\Gamma}\|\nabla\mathcal{L}_s(\boldsymbol{\theta})\| \leq L_s$, where $\Gamma$ contains $\{\boldsymbol{\theta}_k\}_{k=s}^t$ and the line segments between successive iterates.*

*(A2) (Entropy Lipschitzness) For each layer $\ell$, $H(Z_\ell(\boldsymbol{\theta}))$ is Lipschitz in $\boldsymbol{\theta}$ with constant $c_\ell$ along $\Gamma$.*

*(A3) (Local strong convexity/PL for task $k$) The empirical objective $J_k(\boldsymbol{\theta}) := \hat{\mathcal{L}}_k(\boldsymbol{\theta}) = \hat{R}_k(\boldsymbol{\theta}) + \lambda\hat{\Delta}_k(\boldsymbol{\theta})$ is $\mu_k$-strongly convex on the segment between $\boldsymbol{\theta}_{k-1}$ and $\boldsymbol{\theta}_k$, i.e.,*

$$\langle \nabla J_k(\boldsymbol{\theta}) - \nabla J_k(\boldsymbol{\theta}'),\, \boldsymbol{\theta} - \boldsymbol{\theta}'\rangle \geq \mu_k \|\boldsymbol{\theta} - \boldsymbol{\theta}'\|^2.$$

*Then*

$$\mathcal{F}_{s\to t} \leq L_s \sum_{k=s+1}^{t} \frac{1}{\mu_k}\Big(\|\nabla\hat{R}_k(\boldsymbol{\theta}_{k-1})\| + 2\lambda C_\Delta\sqrt{\hat{\Delta}_k(\boldsymbol{\theta}_{k-1})}\Big), \qquad C_\Delta := \Big(\sum_{\ell=1}^{L} c_\ell^2\Big)^{1/2}. \quad (9)$$

*Equivalently, since $\|\nabla\hat{R}_k\| \leq \|\nabla\hat{\mathcal{L}}_k\| + 2\lambda C_\Delta\sqrt{\hat{\Delta}_k}$,*

$$\mathcal{F}_{s\to t} \leq L_s \sum_{k=s+1}^{t} \frac{1}{\mu_k}\Big(\|\nabla\hat{\mathcal{L}}_k(\boldsymbol{\theta}_{k-1})\| + 4\lambda C_\Delta\sqrt{\hat{\Delta}_k(\boldsymbol{\theta}_{k-1})}\Big).$$

Due to space limitations, we provide theorem proof in Appendix.

## 5 EXPERIMENT

### 5.1 EXPERIMENT SETUP

**Datasets**: We evaluate our method on three benchmark datasets: CIFAR-10, CIFAR-100, and Tiny-ImageNet. Following standard class-incremental learning protocols, we partition each dataset into a sequence of distinct tasks. Specifically, CIFAR-10 is divided into 5 tasks of 2 classes each, CIFAR-100 is split into 10 tasks of 10 classes each, and Tiny-ImageNet is partitioned into 10 tasks of 20 classes each.

**Baselines**: We compare our method to strong baselines, including AGEM (Chaudhry et al., 2019a), ER (Chaudhry et al., 2019b), MIR (Aljundi et al., 2019a), GSS (Aljundi et al., 2019b), ASER (Shim et al., 2021), ER-AML (Caccia et al., 2022), GDumb (Prabhu et al., 2020), SCR (Mai et al., 2021), OCM (Guo et al., 2022), OnPro (Wei et al., 2023), GSA (Guo et al., 2023), DER++ (Buzzega et al., 2020), IL2A (Zhu et al., 2021), CO2L (Cha et al., 2021a), LUCIR (Hou et al., 2019), CCIL (Mittal et al., 2021), BIC (Wu et al., 2019), SSIL (Ahn et al., 2021), and MOSE (Yan et al., 2024).

**Implementation details**: For our experiments, baseline results for all methods were adapted from (Yan et al., 2024). The only exception was the MOSE baseline itself, which we reproduced to ensure a fair comparison. On the Tiny-ImageNet dataset, our reproduction using the unmodified official code yielded between 2 to 6% higher accuracy than the results reported in the original paper. We therefore average these baselines for all subsequent comparisons. All experiments were conducted on a single NVIDIA RTX 2080 Ti GPU with 12GB of VRAM, except for Tiny-ImageNet (with a buffer size of 10,000) required an NVIDIA RTX A4000 with 16GB of RAM. Each reported result is the mean and standard deviation computed over 10 independent runs.

### 5.2 RESULTS

Our proposed method demonstrates significant improvements over existing state-of-the-art approaches in continual learning, as shown in Table 1 (overall accuracy) and Table 2 (forgetting). On the Split CIFAR-100 benchmark, our method outperforms the strongest baseline by up to 2.0%

Table 1: Comprehensive comparison of continual learning methods on Split CIFAR-100 and Split Tiny-ImageNet under various memory constraints. All values are Accuracy (%).

| Method | Split CIFAR-100 (10 tasks) - ACC(%) ↑ | | | Split Tiny-ImageNet (100 tasks) - ACC(%) ↑ | | |
| --- | --- | --- | --- | --- | --- | --- |
| | M = 1k | M = 2k | M = 5k | M = 2k | M = 4k | M = 10k |
| AGEM (2019) | 5.8±0.2 | 5.9±0.3 | 6.1±0.4 | 0.9±0.1 | 2.0±0.5 | 3.9±0.2 |
| ER (2019) | 15.7±0.3 | 21.3±0.5 | 28.8±0.8 | 4.7±0.5 | 10.1±0.7 | 11.7±0.2 |
| MIR (2019) | 16.0±0.4 | 19.0±0.1 | 24.1±0.2 | 6.1±0.5 | 11.7±0.2 | 13.5±0.2 |
| GSS (2019) | 11.1±0.2 | 13.3±0.5 | 17.4±0.1 | 3.3±0.5 | 10.0±0.2 | 10.5±0.2 |
| ASER (2021) | 16.4±0.3 | 12.2±1.9 | 27.1±0.3 | 5.3±0.3 | 8.2±0.2 | 10.3±0.4 |
| ER-AML (2022) | 16.1±0.4 | 17.6±0.5 | 22.6±0.1 | 5.4±0.2 | 7.1±0.5 | 10.1±0.4 |
| GDumb (2020) | 17.1±0.4 | 25.1±0.2 | 38.6±0.5 | 12.6±0.1 | 12.7±0.3 | 15.7±0.2 |
| SCR (2021) | 27.3±0.4 | 30.8±0.5 | 36.5±0.3 | 12.6±1.1 | 18.2±0.1 | 21.1±1.1 |
| OCM (2022) | 28.1±0.3 | 35.0±0.4 | 42.4±0.5 | 15.7±0.2 | 21.2±0.4 | 27.0±0.3 |
| OnPro (2023) | 30.0±0.4 | 35.9±0.6 | 41.3±0.5 | 16.9±0.4 | 22.1±0.4 | 29.8±0.5 |
| GSA (2023) | 31.4±0.2 | 39.7±0.6 | 49.7±0.2 | 18.4±0.4 | 26.0±0.2 | 33.2±0.4 |
| DER++ (2020) | 15.3±0.2 | 19.7±1.5 | 27.0±0.7 | 4.5±0.3 | 10.1±0.3 | 17.6±0.5 |
| IL2A (2021) | 18.2±1.2 | 19.7±0.5 | 22.4±0.2 | 5.5±0.7 | 8.1±1.2 | 11.6±0.4 |
| Co2L (2021) | 17.1±0.4 | 24.2±0.2 | 32.2±0.5 | 10.1±0.2 | 15.8±0.4 | 22.5±1.2 |
| LUCIR (2019) | 8.6±1.3 | 19.5±0.7 | 16.9±0.5 | 7.6±0.5 | 9.6±0.7 | 12.5±0.7 |
| CCIL (2021) | 18.5±0.3 | 19.1±0.4 | 20.5±0.3 | 5.6±0.9 | 7.0±0.5 | 15.2±0.5 |
| BiC (2019) | 21.2±0.3 | 36.1±1.3 | 42.5±1.2 | 10.2±0.9 | 18.9±0.3 | 25.2±0.6 |
| SSIL (2021) | 26.0±0.1 | 33.1±0.5 | 39.5±0.4 | 9.6±0.7 | 15.2±1.5 | 21.1±0.1 |
| MOSE (2024) | 37.4±0.3 | 47.0±0.4 | 55.6±0.4 | 24.7±0.5 | 32.4±0.3 | 40.6±0.5 |
| GRACE (Ours) | **39.4±0.4** | **47.6±0.1** | **56.3±0.1** | **28.1±0.2** | **34.8±0.2** | **41.4±0.3** |

in average accuracy, and the performance gains are even more pronounced on the more challenging Split Tiny-ImageNet dataset, where our method achieves up to 3.4% higher average accuracy. When compared to the second-most accurate method, our approach reduces the forgetting metric by an average of 1.1% on CIFAR-100 and 6.6% on Tiny-ImageNet. While some methods focusing exclusively on mitigating forgetting may report lower forgetting values in isolation, they do so at a significant cost to overall accuracy, making our method the most effective and practical solution. We attribute these gains to our dynamic, layer-aware regularization strategy, which contrasts with the static approaches common in prior work. The improvement in overall accuracy is primarily driven by the entropy scaling component. By selectively penalizing over-confident layers, our method effectively mitigates overfitting, a conclusion supported by observing higher validation accuracy despite lower training accuracy compared to baselines. Concurrently, the reduction in catastrophic forgetting stems from the adaptive training mechanism. By constraining updates to well-performing layers, this component successfully preserves previously acquired knowledge. Furthermore, we validate the practical utility of our method in resource-constrained environments in Table 3, demonstrating its strong performance in small-buffer scenarios characteristic of online learning.

## 5.3 ABLATION STUDY

We perform ablation study for GRACE in Table 4.

**w/o Entropy Scaling:** Removing the entropy scaling mechanism causes the most significant performance degradation, with accuracy falling to 37.2%. This confirms that adaptive scaling is the core contribution of our method. The performance suffers because, without scaling, any entropy regularization is applied uniformly. This is detrimental because early layers in a network are responsible for learning general, low-level features (e.g., edges, textures) that are common across many classes.

**w/o Adaptive Training:** Removing the adaptive training component resulted in a modest drop in accuracy from 39.4% to 38.9%. This is expected, as this mechanism is designed to intelligently manage the stability-plasticity trade-off. By adaptively reducing the regularization strength for layers that have already learned robust features for past tasks, it preserves critical knowledge. Removing this targeted intervention leads to slightly increased forgetting and a predictable drop in accuracy.

Table 2: Average Forgetting results on Split CIFAR-100 and Split Tiny-ImageNet benchmarks.

| Method | Split CIFAR-100 (10 tasks) - AF(%) ↓ | | | Split Tiny-ImageNet (100 tasks) - AF(%) ↓ | | |
|---|---|---|---|---|---|---|
| | M = 1k | M = 2k | M = 5k | M = 2k | M = 4k | M = 10k |
| AGEM (2019) | 77.6±2.0 | 76.9±1.5 | 78.3±1.2 | 73.9±0.2 | 77.9±0.2 | 74.1±0.3 |
| ER (2019) | 66.1±1.3 | 59.3±0.9 | 60.0±1.6 | 68.2±2.8 | 66.2±0.8 | 67.2±0.2 |
| MIR (2019) | 24.5±0.3 | 21.4±0.3 | 21.0±0.1 | 61.1±3.2 | 60.4±0.5 | 59.5±0.3 |
| GSS (2019) | 73.4±4.2 | 69.3±3.1 | 70.9±2.9 | 72.8±1.2 | 72.6±0.4 | 71.5±0.2 |
| ASER (2021) | 25.0±0.2 | 12.2±1.9 | 13.2±0.1 | 65.7±0.7 | 64.2±0.2 | 62.2±0.1 |
| ER-AML (2022) | 51.5±0.8 | 49.2±0.5 | 38.7±0.6 | 47.4±0.5 | 43.2±0.3 | 41.0±0.5 |
| GDumb (2020) | 16.7±0.5 | 17.6±0.2 | 16.8±0.4 | 15.9±0.5 | 14.6±0.3 | 11.7±0.2 |
| SCR (2021) | 17.5±0.2 | 11.6±0.5 | 5.6±0.4 | 19.4±0.3 | 15.4±0.3 | 14.9±0.7 |
| OCM (2022) | 12.2±0.3 | 8.5±0.3 | 4.5±0.3 | 23.5±1.9 | 21.0±0.3 | 18.6±0.5 |
| OnPro (2023) | 10.4±0.5 | 6.1±0.6 | 5.3±0.6 | 17.4±0.4 | 16.8±0.4 | 14.6±0.3 |
| GSA (2023) | 33.2±0.6 | 22.8±0.4 | 8.7±0.3 | 35.5±0.3 | 25.8±0.4 | 16.9±0.6 |
| DER++ (2020) | 43.4±0.2 | 44.0±1.9 | 25.8±3.5 | 67.2±1.7 | 63.6±0.3 | 55.2±0.7 |
| IL2A (2021) | 24.6±0.6 | 12.5±0.7 | 20.0±0.5 | 65.5±0.7 | 60.1±0.5 | 57.6±1.1 |
| Co2L (2021) | 16.9±0.4 | 16.6±0.6 | 9.9±0.7 | 60.5±0.5 | 52.5±0.9 | 42.5±0.8 |
| LUCIR (2019) | 60.0±0.1 | 47.5±0.9 | 44.3±0.7 | 46.4±0.7 | 42.2±0.9 | 37.6±0.7 |
| CCIL (2021) | 16.7±0.5 | 16.1±0.3 | 17.5±0.2 | 59.4±0.3 | 56.2±1.3 | 48.9±0.6 |
| BiC (2019) | 40.2±0.4 | 30.9±0.7 | 18.7±0.5 | 43.5±0.5 | 32.9±0.5 | 24.9±0.4 |
| SSIL (2021) | 40.1±0.5 | 33.9±1.2 | 21.7±0.8 | 44.4±0.7 | 36.6±0.7 | 29.0±0.7 |
| MOSE (2024) | 34.7±0.3 | 23.6±0.4 | 12.7±0.4 | 33.3±0.5 | 22.1±0.4 | 11.5±0.4 |
| GRACE (Ours) | 33.9±0.3 | 22.1±0.4 | 11.6±0.5 | 22.7±1.0 | 15.3±0.8 | 8.95±0.3 |

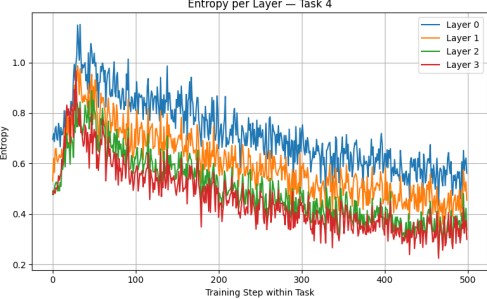

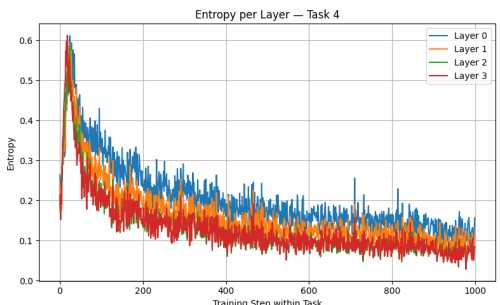

(a) **(Left)** In a baseline model without our intervention, a significant entropy divergence emerges during training. Earlier layers consistently maintain high entropy, while deeper layers collapse to a low-entropy state, suggesting over-confidence.

(b) **(Right)** With entropy scaling, the entropies across all layers are successfully regularized. They converge towards a stable, medium-entropy state, showing that our method prevents individual layers from becoming either over-confident or under-confident.

Table 3: Comparison of memory efficiency on Split CIFAR-100 and Split Tiny-ImageNet.

| Method | Split CIFAR-100 | | Split Tiny-ImageNet | |
|---|---|---|---|---|
| | M=200 | M=500 | M=500 | M=1K |
| OCM (2022) | 12.2±0.4 | 19.7±0.5 | 7.3±0.5 | 10.5±0.6 |
| OnPro (2023) | 14.1±0.9 | 21.5±1.4 | 7.2±0.4 | 10.2±0.3 |
| GSA (2023) | 14.9±0.3 | 22.9±0.2 | 10.4±0.3 | 14.8±0.2 |
| MOSE (2024) | 20.2±0.5 | 28.3±0.7 | 15.2±0.7 | 20.2±0.9 |
| GRACE (Ours) | **21.5±0.5** | **29.8±0.6** | **16.6±0.6** | **21.7±0.8** |

Table 4: Ablation study on Split-CIFAR-100 (with a buffer size of 1,000), showing accuracy drop when removing components.

| Method / Variation | Accuracy (%) |
|---|---|
| **Main Model (Full)** | **39.4** |
| w/o Entropy Scaling | 37.2 |
| w/o Adaptive Training | 38.9 |

# 6 CONCLUSION

In this work, we introduced a novel, layer-wise framework to mitigate catastrophic forgetting in continual learning. Our core method dynamically regularizes each layer by applying a penalty inversely proportional to its output entropy. Our approach is principled and modular and can be readily integrated into existing continual learning pipelines. We have demonstrated its effectiveness through significant performance gains on the CIFAR-10, CIFAR-100, and Tiny-ImageNet benchmarks.

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

## A APPENDIX

**Theorem A.1** (PAC-Bayes generalization with entropy control). *Fix $\delta \in (0,1)$. For task $t$ with sample size $n_t \geq 2$, let*

$$\mathcal{L}_t(\boldsymbol{\theta}) = R_t(\boldsymbol{\theta}) + \lambda \Delta_t(\boldsymbol{\theta}), \qquad R_t(\boldsymbol{\theta}) = \mathbb{E}_{(x,y)\sim\mathcal{D}_t}[\ell_t(f_{\boldsymbol{\theta}}(x), y)],$$

*where $\ell_t \in [0,1]$ and $\Delta_t(\boldsymbol{\theta}) = \sum_{\ell=1}^{L} \left( H_l(\boldsymbol{\theta}) - H_{\ell,t}^* \right)^2$.*

*Let $\hat{\mathcal{L}}_t(\boldsymbol{\theta}) = \hat{R}_t(\boldsymbol{\theta}) + \lambda \hat{\Delta}_t(\boldsymbol{\theta})$ be the empirical analogue on $n_t$ samples. For any posterior $q_t$ absolutely continuous w.r.t. a prior $p_t$ (chosen before seeing the task-$t$ data), with probability at least $1 - \delta$ over the sample,*

$$\mathbb{E}_{\boldsymbol{\theta}\sim q_t}\left[\mathcal{L}_t(\boldsymbol{\theta})\right] \leq \mathbb{E}_{\boldsymbol{\theta}\sim q_t}\left[\hat{\mathcal{L}}_t(\boldsymbol{\theta})\right] + \sqrt{\frac{\mathrm{KL}(q_t\|p_t) + \ln\frac{2\sqrt{n_t}}{\delta}}{2(n_t - 1)}} + \lambda \mathbb{E}_{\boldsymbol{\theta}\sim q_t}\left[\Delta_t(\boldsymbol{\theta})\right]. \tag{10}$$

*Moreover, if $p_t = \mathcal{N}(\boldsymbol{\theta}_{t-1}, \Sigma_p)$ and $q_t$ has mean $\bar{\boldsymbol{\theta}}_t$ and covariance $\Sigma_q$,*

$$\mathrm{KL}(q_t\|p_t) \leq \frac{\kappa_t}{2}\|\bar{\boldsymbol{\theta}}_t - \boldsymbol{\theta}_{t-1}\|^2 + C_t, \qquad \kappa_t := \lambda_{\max}(\Sigma_p^{-1}), \quad C_t := \tfrac{1}{2}\left(\mathrm{tr}(\Sigma_p^{-1}\Sigma_q) - k + \ln\frac{\det\Sigma_p}{\det\Sigma_q}\right).$$

*Proof of Theorem A.1.* By a standard PAC-Bayes inequality for $[0,1]$-bounded losses (e.g., Seeger/McAllester form), with probability at least $1 - \delta$ over the sample of size $n_t \geq 2$, for any posterior $q_t \ll p_t$,

$$\mathbb{E}_{\boldsymbol{\theta}\sim q_t}[R_t(\boldsymbol{\theta})] \leq \mathbb{E}_{\boldsymbol{\theta}\sim q_t}[\hat{R}_t(\boldsymbol{\theta})] + \sqrt{\frac{\mathrm{KL}(q_t\|p_t) + \ln\frac{2\sqrt{n_t}}{\delta}}{2(n_t - 1)}}. \tag{1}$$

A succinct derivation is as follows. Let $r_{\boldsymbol{\theta}} = \hat{R}_t(\boldsymbol{\theta})$ and $R_{\boldsymbol{\theta}} = R_t(\boldsymbol{\theta})$. The "shift-of-measure" (Donsker–Varadhan) inequality implies that for any measurable $\phi$,

$$\mathbb{E}_{q_t}[\phi(\boldsymbol{\theta})] \leq \mathrm{KL}(q_t\|p_t) + \ln\mathbb{E}_{p_t}[e^{\phi(\boldsymbol{\theta})}].$$

Apply this with $\phi(\boldsymbol{\theta}) = \lambda(R_{\boldsymbol{\theta}} - r_{\boldsymbol{\theta}})$ and bound the moment-generating function uniformly over $\boldsymbol{\theta}$ by Hoeffding's lemma for $[0,1]$-bounded losses (together with a union/intersection trick that yields the $n_t - 1$ and the $\ln(2\sqrt{n_t}/\delta)$ refinements), then optimize over $\lambda > 0$ to obtain equation 1.

By definition,

$$\mathbb{E}_{q_t}[\mathcal{L}_t(\boldsymbol{\theta})] = \mathbb{E}_{q_t}[R_t(\boldsymbol{\theta})] + \lambda \mathbb{E}_{q_t}[\Delta_t(\boldsymbol{\theta})].$$

Combining with equation 1 gives

$$\mathbb{E}_{q_t}[\mathcal{L}_t(\boldsymbol{\theta})] \leq \mathbb{E}_{q_t}[\hat{R}_t(\boldsymbol{\theta})] + \sqrt{\frac{\mathrm{KL}(q_t\|p_t) + \ln\frac{2\sqrt{n_t}}{\delta}}{2(n_t - 1)}} + \lambda \mathbb{E}_{q_t}[\Delta_t(\boldsymbol{\theta})]. \tag{2}$$

Now note $\hat{\mathcal{L}}_t(\boldsymbol{\theta}) = \hat{R}_t(\boldsymbol{\theta}) + \lambda \hat{\Delta}_t(\boldsymbol{\theta}) \geq \hat{R}_t(\boldsymbol{\theta})$ since $\hat{\Delta}_t(\boldsymbol{\theta}) \geq 0$. Hence

$$\mathbb{E}_{q_t}[\hat{R}_t(\boldsymbol{\theta})] \leq \mathbb{E}_{q_t}[\hat{\mathcal{L}}_t(\boldsymbol{\theta})],$$

and substituting this into equation 2 yields the claimed bound equation 10. (This step deliberately avoids an empirical-process bound for $\Delta_t - \hat{\Delta}_t$; adding such a bound would replace the last $+\lambda\mathbb{E}_{q_t}[\Delta_t]$ term by $+\lambda\mathbb{E}_{q_t}[\hat{\Delta}_t] +$ a vanishing $O_{\mathbb{P}}(1/\sqrt{n_t})$ term.)

If $p_t = \mathcal{N}(\boldsymbol{\theta}_{t-1}, \Sigma_p)$ and $q_t$ has mean $\bar{\boldsymbol{\theta}}_t$ and covariance $\Sigma_q$, the Gaussian KL identity gives

$$\mathrm{KL}(q_t\|p_t) = \tfrac{1}{2}\left(\mathrm{tr}(\Sigma_p^{-1}\Sigma_q) + (\bar{\boldsymbol{\theta}}_t - \boldsymbol{\theta}_{t-1})^\top\Sigma_p^{-1}(\bar{\boldsymbol{\theta}}_t - \boldsymbol{\theta}_{t-1}) - k + \ln\frac{\det\Sigma_p}{\det\Sigma_q}\right).$$

Using $(\bar{\boldsymbol{\theta}}_t - \boldsymbol{\theta}_{t-1})^\top\Sigma_p^{-1}(\bar{\boldsymbol{\theta}}_t - \boldsymbol{\theta}_{t-1}) \leq \lambda_{\max}(\Sigma_p^{-1})\|\bar{\boldsymbol{\theta}}_t - \boldsymbol{\theta}_{t-1}\|^2$ gives the stated bound with $\kappa_t = \lambda_{\max}(\Sigma_p^{-1})$ and the remaining terms absorbed into $C_t$. $\qquad\square$

**Theorem A.2** (Forgetting bound via parameter drift). *Let $\mathcal{F}_{s \to t} := \mathcal{L}_s(\boldsymbol{\theta}_t) - \mathcal{L}_s(\boldsymbol{\theta}_s)$ for $1 \leq s < t \leq T$. Assume:*

*(A1) $\ell_s \in [0, 1]$ and along the optimization trajectory the population objective $\mathcal{L}_s$ has bounded gradient: $\sup_{\boldsymbol{\theta} \in \Gamma} \|\nabla \mathcal{L}_s(\boldsymbol{\theta})\| \leq L_s$, where $\Gamma$ contains $\{\boldsymbol{\theta}_k\}_{k=s}^t$ and the line segments between successive iterates.*

*(A2) (Entropy Lipschitzness) For each layer $\ell$, $H_l(\boldsymbol{\theta})$ is Lipschitz in $\boldsymbol{\theta}$ with constant $c_\ell$ along $\Gamma$.*

*(A3) (Local strong convexity/PL for task $k$) The empirical objective $J_k(\boldsymbol{\theta}) := \hat{\mathcal{L}}_k(\boldsymbol{\theta}) = \hat{R}_k(\boldsymbol{\theta}) + \lambda \hat{\Delta}_k(\boldsymbol{\theta})$ is $\mu_k$-strongly convex on the segment between $\boldsymbol{\theta}_{k-1}$ and $\boldsymbol{\theta}_k$, i.e.,*

$$\langle \nabla J_k(\boldsymbol{\theta}) - \nabla J_k(\boldsymbol{\theta}'), \, \boldsymbol{\theta} - \boldsymbol{\theta}' \rangle \geq \mu_k \|\boldsymbol{\theta} - \boldsymbol{\theta}'\|^2.$$

*Then*

$$\mathcal{F}_{s \to t} \leq L_s \sum_{k=s+1}^{t} \frac{1}{\mu_k} \Big( \|\nabla \hat{R}_k(\boldsymbol{\theta}_{k-1})\| + 2\lambda \, C_\Delta \sqrt{\hat{\Delta}_k(\boldsymbol{\theta}_{k-1})} \Big), \qquad C_\Delta := \Big( \sum_{\ell=1}^{L} c_\ell^2 \Big)^{1/2}. \quad (11)$$

*Equivalently, since $\|\nabla \hat{R}_k\| \leq \|\nabla \hat{\mathcal{L}}_k\| + 2\lambda C_\Delta \sqrt{\hat{\Delta}_k}$,*

$$\mathcal{F}_{s \to t} \leq L_s \sum_{k=s+1}^{t} \frac{1}{\mu_k} \Big( \|\nabla \hat{\mathcal{L}}_k(\boldsymbol{\theta}_{k-1})\| + 4\lambda \, C_\Delta \sqrt{\hat{\Delta}_k(\boldsymbol{\theta}_{k-1})} \Big).$$

*Proof of Theorem A.2.* **Reduce forgetting to parameter displacement.** By the fundamental theorem of calculus along the line segment from $\boldsymbol{\theta}_s$ to $\boldsymbol{\theta}_t$ and the bounded-gradient assumption (A1),

$$\mathcal{F}_{s \to t} = \int_0^1 \langle \nabla \mathcal{L}_s(\boldsymbol{\theta}_s + \tau(\boldsymbol{\theta}_t - \boldsymbol{\theta}_s)), \, \boldsymbol{\theta}_t - \boldsymbol{\theta}_s \rangle \, d\tau \leq \Big( \sup_{\boldsymbol{\theta} \in \Gamma} \|\nabla \mathcal{L}_s(\boldsymbol{\theta})\| \Big) \|\boldsymbol{\theta}_t - \boldsymbol{\theta}_s\| \leq L_s \|\boldsymbol{\theta}_t - \boldsymbol{\theta}_s\|.$$

By the triangle inequality,

$$\|\boldsymbol{\theta}_t - \boldsymbol{\theta}_s\| \leq \sum_{k=s+1}^{t} \|\boldsymbol{\theta}_k - \boldsymbol{\theta}_{k-1}\|.$$

Hence

$$\mathcal{F}_{s \to t} \leq L_s \sum_{k=s+1}^{t} \|\boldsymbol{\theta}_k - \boldsymbol{\theta}_{k-1}\|. \tag{12}$$

**Bound each inter-task jump by the local geometry of $J_k$.** Since $\boldsymbol{\theta}_k$ is a (local) minimizer or a first-order stationary point of $J_k$ on task $k$, $\nabla J_k(\boldsymbol{\theta}_k) = 0$. By strong convexity/strong monotonicity along the segment (Assumption (A3)) and Cauchy–Schwarz,

$$\mu_k \|\boldsymbol{\theta}_k - \boldsymbol{\theta}_{k-1}\| \leq \|\nabla J_k(\boldsymbol{\theta}_{k-1}) - \nabla J_k(\boldsymbol{\theta}_k)\| = \|\nabla J_k(\boldsymbol{\theta}_{k-1})\|.$$

Therefore

$$\|\boldsymbol{\theta}_k - \boldsymbol{\theta}_{k-1}\| \leq \frac{1}{\mu_k} \|\nabla J_k(\boldsymbol{\theta}_{k-1})\| = \frac{1}{\mu_k} \|\nabla \hat{R}_k(\boldsymbol{\theta}_{k-1}) + \lambda \nabla \hat{\Delta}_k(\boldsymbol{\theta}_{k-1})\|. \tag{13}$$

**Control the entropy-gradient by entropy deviation.** Write $\hat{\Delta}_k(\boldsymbol{\theta}) = \sum_{\ell=1}^{L} d_{\ell,k}(\boldsymbol{\theta})^2$ with $d_{\ell,k}(\boldsymbol{\theta}) = H_l(\boldsymbol{\theta})) - H_{\ell,k}^*$.

By the chain rule,

$$\nabla \hat{\Delta}_k(\boldsymbol{\theta}) = 2 \sum_{\ell=1}^{L} d_{\ell,k}(\boldsymbol{\theta}) \, \nabla H_l(\boldsymbol{\theta})).$$

By Assumption (A2) and Rademacher's theorem (Lipschitz $\Rightarrow$ a.e. differentiable with gradient norm bounded by the Lipschitz constant) we have $\|\nabla H_l(\boldsymbol{\theta})\| \leq c_\ell$ along $\Gamma$. Thus, by Cauchy–Schwarz,

$$\|\nabla\hat{\Delta}_k(\boldsymbol{\theta})\| \;\leq\; 2\Big(\sum_{\ell=1}^{L} d_{\ell,k}(\boldsymbol{\theta})^2\Big)^{1/2}\Big(\sum_{\ell=1}^{L} c_\ell^2\Big)^{1/2} \;=\; 2\,C_\Delta\,\sqrt{\hat{\Delta}_k(\boldsymbol{\theta})}. \tag{14}$$

Evaluating at $\boldsymbol{\theta} = \boldsymbol{\theta}_{k-1}$ yields

$$\big\|\nabla J_k(\boldsymbol{\theta}_{k-1})\big\| \;\leq\; \big\|\nabla\hat{R}_k(\boldsymbol{\theta}_{k-1})\big\| \;+\; 2\lambda\,C_\Delta\,\sqrt{\hat{\Delta}_k(\boldsymbol{\theta}_{k-1})}.$$

Combine this estimate with equation 13 to obtain

$$\|\boldsymbol{\theta}_k - \boldsymbol{\theta}_{k-1}\| \;\leq\; \frac{1}{\mu_k}\Big(\big\|\nabla\hat{R}_k(\boldsymbol{\theta}_{k-1})\big\| \;+\; 2\lambda\,C_\Delta\,\sqrt{\hat{\Delta}}_k(\boldsymbol{\theta}_{k-1})\Big).$$

Plug the last inequality into equation 12 to conclude

$$\mathcal{F}_{s\to t} \;\leq\; L_s \sum_{k=s+1}^{t} \frac{1}{\mu_k}\Big(\big\|\nabla\hat{R}_k(\boldsymbol{\theta}_{k-1})\big\| \;+\; 2\lambda\,C_\Delta\,\sqrt{\hat{\Delta}}_k(\boldsymbol{\theta}_{k-1})\Big),$$

which is equation 11. Finally, since $\|\nabla\hat{R}_k(\boldsymbol{\theta}_{k-1})\| \leq \|\nabla\hat{\mathcal{L}}_k(\boldsymbol{\theta}_{k-1})\| + \lambda\|\nabla\hat{\Delta}_k(\boldsymbol{\theta}_{k-1})\| \leq \|\nabla\hat{\mathcal{L}}_k(\boldsymbol{\theta}_{k-1})\| + 2\lambda C_\Delta\sqrt{\hat{\Delta}_k(\boldsymbol{\theta}_{k-1})}$ by equation 14, the equivalent variant stated in the theorem also follows. $\qquad\square$

## B  BAYESIAN DERIVATION OF ADAPTIVE ENTROPY SCALING

We model the adaptive entropy scaling factor $\gamma_\ell$ in layer $\ell$ as a latent variable in a Bayesian framework. The goal is to infer a suitable regularization strength for each layer, based on its entropy $H_\ell$. We use variational inference to approximate the posterior distribution over $\gamma_\ell$ given the observed entropy.

### B.1  GENERATIVE MODEL

Let $\gamma_\ell \in \mathbb{R}_+$ be the entropy regularization strength for layer $\ell$, and $H_\ell \in \mathbb{R}_+$ the entropy observed at that layer. We assume a Gaussian noise model for entropy given regularization:

$$H_\ell = H^* + \frac{c}{\gamma_\ell} + \varepsilon_\ell, \quad \varepsilon_\ell \sim \mathcal{N}(0, \sigma^2)$$

Thus, the likelihood becomes:

$$p(H_\ell \mid \gamma_\ell) = \mathcal{N}\left(H_\ell \mid H^* + \frac{c}{\gamma_\ell}, \sigma^2\right)$$

We place a log-normal prior on $\gamma_\ell$ to enforce positivity:

$$p(\gamma_\ell) = \text{LogNormal}(\mu_0, \tau^2) \quad \Rightarrow \quad \log\gamma_\ell \sim \mathcal{N}(\mu_0, \tau^2)$$

### B.2  POSTERIOR DISTRIBUTION

The posterior over $\gamma_\ell$ is intractable, so we approximate it via variational inference. Let the variational posterior be:

$$q_\phi(\gamma_\ell) = \text{LogNormal}(\mu_\phi, \sigma_\phi^2) \quad \Rightarrow \quad \log\gamma_\ell \sim \mathcal{N}(\mu_\phi, \sigma_\phi^2)$$

We optimize the evidence lower bound (ELBO):

$$\mathcal{L}(\phi) = \mathbb{E}_{\gamma_\ell \sim q_\phi}\left[\log p(H_\ell \mid \gamma_\ell) + \log p(\gamma_\ell) - \log q_\phi(\gamma_\ell)\right]$$

We place a log-normal prior on $\gamma_\ell$ to ensure positivity and induce regularization:

$$\gamma_\ell \sim \text{LogNormal}(\mu_0, \tau^2) \quad \text{which implies} \quad \log\gamma_\ell \sim \mathcal{N}(\mu_0, \tau^2)$$

The density of the log-normal distribution is:

$$p(\gamma_\ell) = \frac{1}{\gamma_\ell \tau \sqrt{2\pi}} \exp\left(-\frac{(\log \gamma_\ell - \mu_0)^2}{2\tau^2}\right)$$

Taking the logarithm gives the log-prior:

$$\log p(\gamma_\ell) = -\frac{(\log \gamma_\ell - \mu_0)^2}{2\tau^2} - \log \gamma_\ell + \text{const}$$

Substituting the log-densities:

$$\log p(H_\ell \mid \gamma_\ell) = -\frac{1}{2\sigma^2}\left(H_\ell - H^* - \frac{c}{\gamma_\ell}\right)^2 + \text{const}$$

$$\log p(\gamma_\ell) = -\frac{1}{2\tau^2}(\log \gamma_\ell - \mu_0)^2 - \log \gamma_\ell + \text{const}$$

$$\log q_\phi(\gamma_\ell) = -\frac{1}{2\sigma_\phi^2}(\log \gamma_\ell - \mu_\phi)^2 - \log \gamma_\ell + \text{const}$$

Therefore, the ELBO becomes:

$$\mathcal{L}(\phi) = \mathbb{E}_{\gamma_\ell \sim q_\phi}\left[ -\frac{1}{2\sigma^2}\left(H_\ell - H^* - \frac{c}{\gamma_\ell}\right)^2 \right.$$
$$-\frac{1}{2\tau^2}(\log \gamma_\ell - \mu_0)^2 - \log \gamma_\ell$$
$$\left. +\frac{1}{2\sigma_\phi^2}(\log \gamma_\ell - \mu_\phi)^2 + \log \gamma_\ell \right] + \text{const}$$

Observe that the $-\log \gamma_\ell + \log \gamma_\ell$ terms cancel out, simplifying the ELBO.

### B.3 REPARAMETERIZATION TRICK

We use the reparameterization trick for gradient estimation:

$$\log \gamma_\ell = \mu_\phi + \sigma_\phi \cdot \epsilon, \quad \epsilon \sim \mathcal{N}(0, 1)$$

We can now estimate the ELBO and its gradient using Monte Carlo samples of $\epsilon$.

### B.4 DETERMINISTIC APPROXIMATION: GRACE SCALING RULE

To avoid optimizing $\mathcal{L}(\phi)$ explicitly during training, we approximate the *posterior mean*:

$$\hat{\gamma}_\ell = \mathbb{E}_{q_\phi}[\gamma_\ell] = \exp\left(\mu_\phi + \frac{\sigma_\phi^2}{2}\right)$$

Assuming a small variance $\sigma_\phi^2 \approx 0$, we approximate:

$$\hat{\gamma}_\ell \approx \exp(\mu_\phi)$$

Empirically, we set:

$$\mu_\phi \approx \tanh\left(z_\ell\right), \quad \text{where } z_\ell = \frac{H_\ell - \mu_H}{\sigma_H}$$

This yields the final approximation used in GRACE:

$$\gamma_\ell \approx \exp\left(\tanh\left(\frac{H_\ell - \mu_H}{\sigma_H}\right)\right)$$

