# OpenReview forum: "Layer-Wise Feedback Signals: Dynamic Regulation for Continual Learning"
_ICLR.cc/2026/Conference — ICLR 2026 Conference Withdrawn Submission_

### Official Review · Reviewer_JHYd · 2025-10-20

**Soundness:** 2
**Presentation:** 2
**Contribution:** 3
**Rating:** 4
**Confidence:** 4

**Summary:**

The paper introduces GRACE, a layer-wise, entropy-aware framework for continual learning. The method applies a per-layer scaling based on the entropy of intermediate representations and modulates the plasticity of each layer based on its performance on previous tasks. Experiments on standard continual learning benchmarks demonstrate that GRACE improves both average accuracy and forgetting compared to prior baselines.

**Strengths:**

The paper proposes a simple yet effective per-layer feedback mechanism that adaptively regulates learning dynamics across layers, resulting in improved performance and reduced forgetting. The empirical evaluation is comprehensive and it shows how GRACE outperforms previous baselines in two benchmarks. In addition, the theoretical analysis is detailed and it helps justifying the approach.

**Weaknesses:**

- The algorithm includes a per-layer evaluate accuracy (line 6 in Algorithm 1), but the paper does not fully explain how this is implemented. For example, do all layers share the same classifier head?
- Some experimental details are missing, such as the architecture used and the specific hyperparameters selected for training. The details are necessary for reproducibility.
- The ablation experiments report only the accuracy metric, yet the adaptive training component is motivated specifically to reduce forgetting. The addition of forgetting is necessary to support this claim.
- While the method is well described, the experimental section is limited. The tables quantitatively show that GRACE outperforms previous baselines, but it would be interesting also to include a qualitative or quantitative analysis of which layers are more strongly regularized, whether a pattern can be identified and how this approach can be applied to different architectures.
- Minor issue: the figures are not referenced in the main text.

**Questions:**

- The entropy-based regularization aims to encourage layers to learn more general representations by penalizing both overconfident (low-entropy) and underconfident (high-entropy) predictions. Is it always beneficial for all layers to maintain similar entropy levels? Could this discourage desirable layer specialization?
- How sensitive is the method to batch size when computing the z-score for entropy scaling?
- What is the computational overhead (in terms of memory and training time) introduced by the per-layer entropy and accuracy computations?
- Is the approach proposed layer-agnostic? Or in other words can it be applied to different layers' type such as linear, convolutional, attention layers, etc.?
- In the experimental tables, what does M represent?
- What are the main limitations of the proposed method, and what directions for future work do the authors foresee?

---

### Official Review · Reviewer_ueDY · 2025-10-21

**Soundness:** 3
**Presentation:** 1
**Contribution:** 1
**Rating:** 2
**Confidence:** 4

**Summary:**

Continual learning (CL) deals with sequentially training a model on a stream of not-fully-storable data chunks. The central challenge is catastrophic forgetting, caused by overwriting old information learned from prior tasks.
This leads to a stability (not forgetting) vs plasticity (learning new things) tradeoff. In their paper, authors study the contribution of individual layers towards the general performance. They propose a layerwise adaptation regime where they scale the updates to individual layers based on their entropy. Layers with low entropy (highly confident) are nudged to learn more general representations. Layers with high entropy are not modified much. Additionally, authors use adaptive training that quantifies a layer's relative importance for performance on historic tasks. The paper rounds of with a comprehensive set of experiments.

**Strengths:**

Strengths
+ I enjoyed the idea
+ extensive experiments (though see weaknesses)
+ detailed proofs of the entropy regulation

**Weaknesses:**

Weaknesses:
- Paper is missing details that help me understand it: Which layers are regularized (all?)? How is the intermediate head trained (trained anew each time?)? Is normal cross-entropy training included? Do you use pre-trained networks? Hparams optimized? How is forgetting computed (is that BWT or Forgetting Measure)? Are you using an offline or online setting?
- results in Table1 and Table2 not comparable. The extensive experiments rely on results reported by [1] (which is fine by itself and common practice). However, these results are partially from an online (one pass over the data only) setting (!), which is not fairly comparable to the (assumed) offline setting (multiple epochs per task allows) in your work
- No runtime assessments conducted. I think that iterating through all layers twice (as in Algorithm 1) considerably scales runtimes
- Related work missing sections on important fields necessary to assess the paper. E.g., entropy usage in machine learning training, layerwise (pre-)training regimes, and layerwise importances (you already refer to the seminal "Are all layers created equal?" paper, so this could be a starter for further related work into this area)

[1] Yan, H., Wang, L., Ma, K., & Zhong, Y. (2024). Orchestrate latent expertise: Advancing online continual learning with multi-level supervision and reverse self-distillation. In Proceedings of the IEEE/CVF Conference on Computer Vision and Pattern Recognition (pp. 23670-23680).

**Questions:**

Suggestions/Questions
- Make Figure 1 larger
- ALg1, line 27: what is H_l? the non-averaged representation?
- Table1: TinyImageNet does not have 100 tasks, or?
- Am I correct that no "normal training" is done? It seems to be missing from Alg1 at least

---

### Official Review · Reviewer_G6jy · 2025-10-30

**Soundness:** 3
**Presentation:** 2
**Contribution:** 2
**Rating:** 4
**Confidence:** 4

**Summary:**

The paper introduces a continual learning framework, GRACE (Guided Entropy-Adaptive Feedback for Continual Learning), which dynamically regulates learning at the layer level via two mechanisms: Self-Adaptive Entropy Scaling and Adaptive Training. The former applies layer-specific regularization by adjusting feature representations based on output entropy to encourage more generalizable representations; the latter modulates each layer’s contribution according to its performance on previously learned tasks, aiming to preserve prior knowledge while enabling adaptation to new tasks.

**Strengths:**

- The paper addresses a central challenge in continual learning, by regulating the stability-plasticity trade-off at the layer level. Using entropy as a dynamic feedback signal per layer is well-motivated and appears novel in this context.

- The theoretical analysis provides a solid foundation: beyond PAC-Bayes–style generalization arguments and a forgetting bound, the analysis helps justify the design choices and clarifies the method’s expected behavior. The explicit algorithmic specification makes the workflow transparent and facilitates its understanding.

**Weaknesses:**

- The experimental section omits key training details (e.g., number of epochs per task, training schedule, optimizer and hyperparameters), hindering reproducibility and making it difficult to assess fairness in comparisons. Some baseline results appear lower than commonly reported in the literature; without fuller clarification of the training protocol, it is unclear whether this discrepancy stems from implementation choices, hyperparameter settings, or dataset/task configurations. Notice: CIFAR-10 is listed as a benchmark, but no results are reported.

- The method introduces several hyperparameters (e.g., entropy scaling coefficients, target entropy), yet no sensitivity/robustness analysis is provided, leaving unclear how dependent the gains are on specific settings.

- Evaluation is restricted to Split CIFAR-100 and Tiny-ImageNet, and the ablation study does not sufficiently disentangle the contributions of the two components to determine whether improvements are synergistic or merely additive. Moreover, ablation results are presented without variance and appear single-run, weakening statistical confidence.

- The paper does not quantify computational overhead (e.g., wall-clock time, FLOPs, parameter growth). Given the additional per-layer heads, per-batch entropy computations, and inter-task validation passes, the method likely incurs non-negligible training-time cost. Reporting such metrics would clarify practical trade-offs.

**Questions:**

- How sensitive is the proposed method to the entropy-related hyperparameters (e.g., scaling coefficients, target entropy), and has any sensitivity or robustness analysis been performed?

- To what extent can the roles of the entropy-scaling and layer-adaptive components be disentangled?

- It would be useful to report training-time cost, FLOPs, and parameter overhead to clarify the practical trade-offs involved.

---

### Official Review · Reviewer_oJLf · 2025-11-02

**Soundness:** 2
**Presentation:** 3
**Contribution:** 2
**Rating:** 2
**Confidence:** 4

**Summary:**

The paper proposes a *Self-Adaptive Entropy Scaling* approach to mitigate catastrophic forgetting in continual learning. The method adaptively modulates layers within a neural network based on entropy, assigning different levels of regularization according to layer-wise entropy values.

**Strengths:**

- The paper is well-organized and clearly written.

- Theoretical analysis and empirical results are well-balanced.

- The idea of addressing catastrophic forgetting at the layer level is novel and conceptually interesting.

**Weaknesses:**

- Several implementation details are insufficiently explained, making it difficult to reproduce or fully evaluate the proposed approach.

- The justification for the proposed entropy-based scaling remains weak. The method does not consider the inherent structural differences between layers (e.g., early layers learning general representations vs. later layers capturing task-specific features). Enforcing similar entropy behavior across all layers might disrupt natural learning dynamics.

- The scalability and computational efficiency of assigning separate classification heads per block or layer are not convincingly addressed.

Although the paper presents an interesting perspective on catastrophic forgetting, the methodological justification is not sufficiently rigorous, and important implementation and efficiency aspects remain unclear. The lack of analysis regarding layer-wise structural differences and entropy interpretation further weakens the paper’s overall contribution.

**Questions:**

1. Could you provide additional analysis on how structural differences between layers affect the proposed entropy scaling?

2. How is performance from previous tasks measured at the layer level? Figure 1 mentions separate classification heads per block, but since the scaling is applied per layer, it seems computationally inefficient. Please clarify how this is implemented and discuss its computational complexity.

3. How do you define the entropy threshold for determining whether a layer is over-confident? The paper states that “We do not need to estimate H∗ since our approach can normalize the entropy values across different layers without knowing H∗”, but the normalization process is not clearly described.

4. The method modulates regularization based on relative entropy within a mini-batch (“our method dynamically modulates the regularization penalty based on a layer’s relative entropy compared to other layers within the same mini-batch, rather than its absolute entropy.”). This implies that batch composition could affect the outcome. How does the method behave under different batch sizes or distributions?

---

### Note · Authors · 2025-11-14

I have read and agree with the venue's withdrawal policy on behalf of myself and my co-authors.